# Characteristic of K3 (CpG-ODN) as a Transcutaneous Vaccine Formulation Adjuvant

**DOI:** 10.3390/pharmaceutics12030267

**Published:** 2020-03-15

**Authors:** Sayami Ito, Sachiko Hirobe, Takuto Kawakita, Mio Saito, Ying-Shu Quan, Fumio Kamiyama, Ken J. Ishii, Mizuho Nagao, Takao Fujisawa, Masashi Tachibana, Naoki Okada

**Affiliations:** 1Project for Vaccine and Immune Regulation, Graduate School of Pharmaceutical Sciences, Osaka University, 1-6 Yamadaoka, Suita, Osaka 565-0871, Japan; ito-sa@phs.osaka-u.ac.jp (S.I.); kawakita-t@phs.osaka-u.ac.jp (T.K.); tacci@phs.osaka-u.ac.jp (M.T.); 2Advanced Research of Medical and Pharmaceutical Sciences, Graduate School of Pharmaceutical Sciences, Osaka University, 1-6 Yamadaoka, Suita, Osaka 565-0871, Japan; hirobe-s@phs.osaka-u.ac.jp; 3Project of Clinical Pharmacology and Therapeutics, Center for Advanced Education and Research in Pharmaceutical Sciences, Graduate School of Pharmaceutical Sciences, Osaka University, 1-6 Yamadaoka, Suita, Osaka 565-0871, Japan; 4Department of Molecular Pharmaceutical Science, Graduate School of Medicine, Osaka University, 2-2 Yamadaoka, Suita, Osaka 565-0871, Japan; 5CosMED Pharmaceutical Co. Ltd., 32 Higashikujokawanishi-cho, Minami-ku, Kyoto 601-8014, Japan; saito@cosmed-pharm.co.jp (M.S.); quan@cosmed-pharm.co.jp (Y.-S.Q.); kamiyama@cosmed-pharm.co.jp (F.K.); 6Division of Vaccine Science, The Institute of Medical Science, The University of Tokyo, 4-6-1 Shirokanedai, Minato-ku, Tokyo 108-8639, Japan; kenishii@ims.u-tokyo.ac.jp; 7Allergy Center and Department of Clinical Research, Mie National Hospital, 357 Osato-kubota, Tsu, Mie 514-0125, Japan; watersail711@gmail.com (M.N.); eosinophilosophy@gmail.com (T.F.); 8Laboratory of Vaccine and Immune Regulation (BIKEN), Graduate School of Pharmaceutical Sciences, Osaka University, 1-6 Yamadaoka, Suita, Osaka 565-0871, Japan

**Keywords:** transcutaneous immunization, microneedle, poke-and-patch, skin vaccination, transcutaneous drug delivery, TLR9 ligand, clinical research, safety test

## Abstract

Transcutaneous immunization (TCI) is easy to use, minimally invasive, and has excellent efficacy in vaccines against infections. We focused on toll-like receptor (TLR) ligands as applicable adjuvants for transcutaneous formulations and characterized immune responses. TCI was performed using poke-and-patch methods, in which puncture holes are formed with a polyglycolic acid microneedle on the back skin of mice. Various TLR ligands were applied to the puncture holes and covered with an ovalbumin-loaded hydrophilic gel patch. During the screening process, K3 (CpG-oligonucleotide) successfully produced more antigen-specific antibodies than other TLR ligands and induced T helper (Th) 1-type polarization. Transcutaneously administered K3 was detected in draining lymph nodes and was found to promote B cell activation and differentiation, suggesting a direct transcutaneous adjuvant activity on B cells. Furthermore, a human safety test of K3-loaded self-dissolving microneedles (sdMN) was performed. Although a local skin reaction was observed at the sdMN application site, there was no systemic side reaction. In summary, we report a K3-induced Th1-type immune response that is a promising adjuvant for transcutaneous vaccine formulations using MN and show that K3-loaded sdMN can be safely applied to human skin.

## 1. Introduction

Infectious diseases occur and spread widely and repeatedly worldwide, making them a leading cause of death in developing countries. Also, well-developed transport networks can accelerate the spread of pathogens and thus, people are threatened by global epidemics of emerging and reemerging infectious diseases such as influenza (H1N1, 2009) and Ebola hemorrhagic fever (2014–2016) [1,2]. Therefore, international infectious disease control is an urgent issue and it is necessary to develop safe and effective vaccines. However, most of the vaccines in practical use are injection preparations that require medical personnel for inoculation and expensive cold-chains for manufacturing, transportation, and storage, which are barriers to the use of vaccines in developing countries.

As a solution to these challenges, we have developed novel transcutaneous administration devices such as self-dissolving (sdMN) and biodegradable polyglycolic acid (PGA-MN) microneedles [3,4]. These devices are easy to use, minimally invasive, and may not require a cold-chain handling, suggesting that these devices could promote vaccine availability in developing countries. We have already demonstrated the safety and efficacy of MN loaded or coated with the influenza hemagglutinin (HA) vaccine or tetanus toxoid/diphtheria toxoid (DT) in animal experiments and clinical studies [5,6,7,8]. To put these transcutaneous vaccine formulations into practical use and to spread the vaccine in developing countries, it is necessary to lower the cost by reducing vaccination dose and number of administrations and to induce stable immune responses. As an approach to address these cost-cutting options, we developed an application of an adjuvant that allows for the enhancement and modification of the immune response to transcutaneous vaccine formulations.

Concomitant use of adjuvants in vaccine formulation development has been utilized for a long time. Aluminum salt (Alum) is used as an adjuvant for most injectable vaccines approved in Japan [9]; however, it is difficult to apply Alum to transcutaneous formulations using MN because of its insolubility and cytotoxicity. Therefore, it is necessary to identify candidate substances that exert adjuvant activity during transcutaneous administration. Many adjuvants under development are ligand molecules targeting innate immunity receptors [10]. Functional signaling pathways of toll-like receptors (TLR) are being elucidated in detail by rigorous basic research, and knockout mice are powerful research tools [11,12]. Moreover, it is also clear that TLR-mediated innate immune responses lead to subsequent acquired immune responses, providing a rational for using TLR ligand molecules as an adjuvant.

In this study, we screened TLR ligands as adjuvants for transcutaneous formulations. Focusing on K3, the ligand of TLR9 with the highest ability to induce antibody production, localization in the skin and draining lymph nodes (DLNs; brachial lymph node, axillary lymph node, and inguinal lymph node), and characteristics of the immune response in DLNs and spleen were examined. Furthermore, the K3-loaded sdMN was prepared and a clinical trial was performed to evaluate the safety of K3 in human skin.

## 2. Materials and Methods

### 2.1. Mice

Female C57BL/6, BALB/c mice and hairless mice [Hos: HR-1] in a BALB/c background (7-w-old) were purchased from SLC Inc. (Hamamatsu, Japan) and TLR9-knockout (TLR9-KO) mice in a C57BL/6 background were purchased from OBS Inc. (Kyoto, Japan) [13,14]. B6 Thy1.1 (B6.PL-Thy1^a^/CyJ), B6 CD45.1 (B6.SJL-Ptprca Pepcb/BoyJ), OT-I (C57BL/6-Tg (TcraTcrb) 1100Mjb/J), and OT-II mice (B6.Cg-Tg (TcraTcrb) 425Cbn/J) were purchased from the Jackson Laboratory (Bar Harbor, ME, USA). These mice were maintained and bred in the experimental animal facility at Osaka University. In this study, the care and use of laboratory animals were properly conducted following the guidelines and policies of the Act on Welfare and Management of Animal in Japan. The protocols and procedures were approved by the Animal Care and Use Committee of Osaka University (protocol number: Douyaku 28-6 (2017/11/20)). Animal experiments were conducted under 2% isoflurane anesthesia inhalation.

### 2.2. Preparation of Ovalbumin (OVA)-Loaded Hydrophilic Gel Patch (HG)

The hydrophilic gel (HG) formulations that contained crosslinked HiPAS 10 (CosMED Pharmaceutical Co., Kyoto, Japan), octyldodecyl lactate, glycerin, and sodium hyaluronan in a ratio of 100:45:30:0.2 (*w/w*) were prepared as previously described [15].

Low endotoxin OVA (<1 EU/mg) was purchased from FUJIFILM Wako Pure Chemical Corporation (Osaka, Japan) and dissolved in sterile distilled water. The OVA solution is dropped onto the HG and dried sufficiently to make 10 μg of OVA concentrated on HG surface. When the HG is applied to intact skin, the stratum corneum layer hydrates the gel, which swells due to water absorption from skin transpiration. The OVA diffuses across the stratum corneum as a result of the high concentration gradient at the skin surface [16].

### 2.3. Immunization

The back hair of mice was removed with Epilat (Kracie Holdings, Tokyo, Japan) 48 h before immunization. The hairless skin was punctured with PGA-MN with a 1 cm diameter, 300 μm needle length, and a total of 481 of needles [4]. TLR ligands were adjusted to each concentration with PBS (Table 1), and 5 μL were dropped onto the punctured skin. The scrambled nucleic acid (ATGCACTCTGCAGGCTTCTC) was used as a control for K3 (ATCGACTCTCGAGCGTTCTC) in equal volume as K3 [17]. Ten micrograms OVA-loaded HG was applied to the skin for 24 h (poke-and-patch method; Figure 1A) [18].

### 2.4. Epidermal and Dermal Sheets

The hairless back skin and ear skin were harvested and floated on 0.5% dispase in complete RPMI (cPRMI; 50 μM 2-ME, 10% FBS, 20 mM HEPES, 1 mM sodium pyruvate, 100 unit/mL penicillin, 100 μg/mL streptomycin and 0.25 μg/mL amphotericin B in RPMI1640). After incubation at 37 °C, epidermal and dermal sheets were isolated and immersed in cPRMI containing 0.25% trypsin and 0.025% DNase. After incubation at 37 °C for 5 min, the solution was passed through a 70-μm cell filter to produce an epidermal cell suspension. The dermal sheet was shredded in cRPMI containing 1% collagenase, 0.25% trypsin and 0.025% DNase, and a dermal cell suspension was made with stirring at 37 °C for 30 min.

### 2.5. Flow Cytometry Analysis

The spleens and draining lymph nodes were collected from the immunized mice and separated into single cells. Cell suspensions were incubated with purified-anti CD16/32 antibody (clone 93) for 15 min at 4 °C. Cocktails of cell surface markers were suspended in staining buffer (2% FBS, 0.05% NaN_3_, 1 × PBS) and incubated on ice for 30 min. Intracellular staining was conducted using an intracellular staining kit (BD, Franklin Lakes, NJ, USA) according to the manufacturer instructions. The stained cells were gently suspended in staining buffer after centrifugation at 400× *g* for 5 min at 4 °C. Stained cells were detected with a FACS Canto II (BD), and collected data were analyzed using FlowJo software (BD). The antibody used was described in the Appendix A.

### 2.6. Measurement of OVA-Specific Antibody Titer

Blood was collected from the orbital sinus over time to obtain blood sera. OVA-specific IgG and IgE titers in sera were measured with an enzyme-linked immunosorbent assay (ELISA). OVA-coated ELISA plates were blocked, and then half-fold serial dilutions of sera were added and incubated for 2 h at 25 °C. Horseradish peroxidase (HRP)-conjugated anti-mouse IgG and IgG subclass antibodies (Southern Biotechnology, Birmingham, AL) were used for the detection of mouse OVA-specific IgG and IgG subclass. Biotin-conjugated rat anti-mouse IgE (BD) and streptavidin-HRP (BD) were used for detecting OVA-specific IgE. The TMB substrate was added and the absorbance (OD450 nm and OD655 nm) was measured on a SpectraMax iD5 (Molecular Devices, LLC., San Jose, CA, USA).

For nasal washing, PBS was injected into the nostrils from the nasopharynx of mice and the fluid that flowed out of the nostrils was collected. OVA-specific IgA antibody was detected using an HRP-conjugated anti-mouse IgA antibody (Southern Biotechnology) and the other methods described above.

The OVA-specific antibody titers were expressed as the reciprocal log2 titer of the highest dilution that generated 0.1 absorbance units after subtracting the absorbance of preimmune sera.

### 2.7. Immunofluorescence Staining

Mice were immunized with OVA and AF647-labeled K3 by the poke-and-patch method and the application site of skin and DLNs were collected over time after HG application to observe the localization of OVA and K3. These tissue samples were immersed in 4% PFA/PBS overnight at 4 °C. After a sucrose hydration treatment, the skin and DLNs were frozen in OCT compound (Sakura Finetek, Tokyo, Japan) and cut into 8-μm-thick sections using a cryostat (Leica Biosystems, Buffalo Grove, IL). The sliced sections were blocked and stained with OVA by the indirect method. OVA was stained with purified anti-OVA from abcam (Cambridge, UK) as a primary antibody and Alexa Fluor 546 anti-rabbit IgG from Thermo Fischer scientific as a secondary antibody. Histological examination of the skin samples was performed on frozen sections that were mounted with Prolong Gold antifade reagent with DAPI (Thermo Fisher Scientific, Waltham, MA) and then photographed using fluorescent microscopy (BZ-8000; Keyence, Osaka, Japan).

### 2.8. Cytokine Assay

Spleens were collected from mice 2 weeks after the second immunization and prepared as a cell suspension. The cells were plated into 24-well plates with 2.5 × 10^6^ cells per well, and 1 mg low endotoxin OVA was added before the cells were cultured for 48 h in a 37 °C incubator. IL-2, IL-4, IL-5, IL-9, IL-10, IL-13, IL-17, IFN-γ, and TNF-α were measured from the culture supernatant using a Bio-Plex mouse cytokine assay (Bio-Rad Laboratories, Inc., Hercules, CA, USA) according to the manufacturer’s instructions.

### 2.9. In Vivo Proliferation Assay for OVA-Specific T Cells

The CD4^+^ T cells from OT-II mice and CD8^+^ T cells from OT-I mice were isolated from spleens and DLNs with an auto MACS Pro Separator (Miltenyi Biotec, Bergisch Gladbach, Germany) and a mouse CD4^+^ T Cell Isolation Kit or mouse CD8^+^ T Cell Isolation Kit (Miltenyi Biotec), respectively. The isolated cells were fluorescently labeled with eFluor 670 and intravenously administered to wild type (WT) mice at 3 × 10^6^ cells/head. The next day, mice were used to perform a transcutaneous immunization (TCI) using the poke-and-patch method. DLNs were collected from these mice 4 d after immunization and the proliferation of each transferred cell was detected by flow cytometry using the fluorescence intensity of eFluor 670.

### 2.10. In Vitro Stimulation of B Cells

The DLNs and spleens were collected from WT or TLR9-KO mice, and B cells were negatively isolated using an auto MACS pro Separator and B cell Isolation Kit, respectively. The B cells were suspended in cRPMI and seeded at 1 × 10^6^ cells/well into 24-well plates. A 1 μg/5 μL solution of K3 or the scrambled K3 control was added to each well and cultured in 37 °C incubator. The B cells were collected 6 h later and the expression of the activation markers described above was analyzed by a FACS.

### 2.11. Safety Evaluation for TCI of K3: Clinical Trial (Phase I)

This was a randomized, double-blind, placebo-controlled, phase I study to investigate the safety of K3 administrated through the skin. The trial was conducted at Mie National Hospital in accordance with the Declaration of Helsinki. This study protocol was approved by the ethics committee of Mie National Hospital (No. 31–55).

The inclusion criteria for the study was as follows: healthy male adults aged 20 to 40 years old; written informed consent was obtained from each participant; subjects complied with the requirements during the study; participants received a consultation as specified in the protocol and can report symptoms. The exclusion criteria were as follows: individuals who received K3 within 180 d before the start of the study; individuals with a past of food or drug anaphylaxis; those with a history of serious cardiovascular system, blood, respiratory system, liver, kidney, gastrointestinal system, or neuropsychiatric system diseases; those with a history of Guillain-Barre syndrome or acute disseminated encephalomyelitis; individuals who received a live vaccine within 27 days before the start of the test or inactivated vaccine toxoid within 6 days (calculated from the day of administration); those who received therapeutic blood transfusions or gamma globulin products within 3 months prior to the start of the test (calculated from the day of administration) or those who received high-dose gamma globulin products (200 mg/kg or more) within 6 months (calculated from the day of administration); women who wished to become pregnant by their partner during the test (from the date of first study drug administration to the date of final follow-up) or who were unable to take contraceptives because of the potential for sexual activity during the study; and those who were considered as unsuitable for participation by the research staff.

The subjects received sdMN with or without K3 (200 μg) on the outer skin of the upper arm using the dedicated applicator for 1 h (Figure 1B). The sdMN was applied twice at different locations on days 0 and 21 (Table 2). Before administration of the sdMN, subjects underwent examinations, vital sign recording, blood sampling, and urinalysis. The collected blood was used for general blood tests, biochemical tests, and cytokine production measurements. The same tests were performed on the day after the sdMN administration. The subjects recorded axillary body temperature, local skin reaction, and rational/objective symptoms in diaries. As the primary outcome, safety was assessed by the type, severity, duration, and incidence of adverse events and adverse reactions that occurred after the first immunization with the sdMN and up to the date of the final follow-up, which will be determined.

### 2.12. Statistical Analysis

Data are expressed as mean ± standard error of the mean (SEM) of results from three to seven mice. Statistical analysis was performed using a Student *t*-Test. In all cases, *p* < 0.05 (*), *p* < 0.01 (**), *p* < 0.005 (***), and *p* < 0.001 (****) were considered statistically significant and highly significant, respectively.

## 3. Results

### 3.1. Screening for Effective Adjuvants for TCI Using sdMN

Analysis of TLRs in mouse skin cells indicated the expression of all TLRs from TLR1 to TLR9 in Langerhans cells (LCs) and dermal dendritic cells (dDCs) functioning as antigen-presenting cells (Figure 2A), while slight expression of TLR3, 7, 8 and 9 were observed in keratinocytes. These results suggested that the immune cells and keratinocytes present in the skin were involved in the induction of innate immunity by detecting the invasion of foreign substances via TLRs. Thus, an excellent adjuvant effect can be expected in these skin cells when TCI is performed using TLR ligand expression.

To evaluate the adjuvant effect of TLR ligands, we transcutaneously delivered OVA and various TLR ligands to the skin using poke-and-patch methods for TCI using PGA-MN. The dose of TLR ligand used in the present study was set with reference to the previously injection dose administration and maximum solubility reported [19,20,21]. An increase in the OVA-specific IgG antibody titer was detected from initial immunization only when used with K3 in two mouse strains (Figure 2B). In the K3 combination group, the OVA-specific IgG antibody titer after multiple immunizations was significantly higher than that of the control and other TLR ligand combination groups. Other adjuvant combination groups showed almost the same antibody titer as the control group, suggesting that it was difficult to induce adjuvant activity with the poke-and-patch method and TLR ligand dose. In the control group, the Th2-type IgG subclass antibody was induced dominantly, indicating that it induced a Th2-type immune response (Figure 2C). However, the Th1-type IgG subclass antibodies such as IgG2c and IgG2a were significantly high in the K3 combination group, suggesting K3 could also induce a Th1-type immune response. The vaccine side effect and OVA-specific IgE antibody titer did not increase in the adjuvant combination group. Specifically, the K3 combination group had similar effects to the control group, indicating that side effects were minimal even when K3 was transcutaneously administered (Figure 2D). OVA-specific CD8^+^ T cells (cytotoxic T lymphocytes) were not produced even with K3, indicating that the induction of cellular immunity was difficult (data not shown). Furthermore, induction of an immune response on the mucosal surface, which is the site of the initial infection of the pathogen, can protect the pathogen invasion. To evaluate the induction of mucosal immunity in the nasal mucosa, the specific IgA antibody titer in the nasal washings of mice was measured, which was below the detection limit, indicating that mucosal immunity was not induced by K3 (data not shown).

To investigate whether the adjuvant activity of transcutaneously administered K3 was induced via TLR9, TCI was performed on hairless back skin of WT and TLR9-KO mice using the poke-and-patch method. WT mice showed an excellent immune response in the K3 combination group, whereas the OVA-specific antibody titer in TLR9-KO mice was not different from the scrambled K3 control combination group (Figure 2E). These results suggested that TCI in combination with K3 could induce not only a rapid and strong antigen-specific immune response through TLR9 stimulation, but also Th1-type immune responses.

### 3.2. Localization of OVA and K3 in Skin and DLNs

To observe the dynamics of OVA and K3 administrated by the poke-and-patch method, the skin where the HG was applied and DLNs were collected at 1, 3, 6, 12, 24, and 48 h and evaluated by immunofluorescent staining. OVA was weakly expressed in the stratum corneum 1–6 h after application of the HG patch and was hardly detected in the skin and DLNs thereafter (Figure 3). The OVA concentrated on the HG surface was either slowly transferred to the stratum corneum or the diffusion efficiency between the HG surface and the skin was low; therefore, OVA could not be detected in the skin. By contrast, K3 was detected at a portion of the skin with puncture holes because of missing a part of the basal cell layer. The K3 diffused into the skin over time and was delivered through lymphatic vessels to DLNs. In DLNs, K3 was observed approximately 12 h after application. Together, these results suggest that the dynamics of OVA-loaded HG and K3 were different and that K3 was transported into the DLNs where it might act on immune cells.

### 3.3. Activation and Differentiation of T cells and B cells after Combination Treatment with K3

To investigate changes in the DLNs and spleen after transcutaneous administration of K3, the number and differentiation status of each immune cell were analyzed by flow cytometry. In the DLNs of the immunization site, the total cells (Figure 4A), T cells (Figure 4B), and B cells significantly increased with K3 combination treatment (Figure 4C). The differentiation of activated CD4^+^ T cells (CD4^+^ CD44^high^) by transcutaneous vaccine combined with K3 was also examined to investigate populations of effector T cells (T_EFF_; CD44^high^, CD127^+^, CD62L^−^), effector memory T cells (T_EM_; CD44^high^, CD127^−^, CD62L^−^) and central memory T cells (T_CM_; CD44^high^, CD127^+^, CD62L^+^) using surface markers. The subpopulation of each CD4^+^ T cell increased; in the CD4^+^ subpopulation, T_EFF_ cells and memory T cells significantly increased with K3 combination treatment compared with the scrambled control. When the subpopulation was analyzed for B cells, germinal center (GC) B cells (CD45R^+^, IgG1^+^, GL7^+^, CD95^+^), memory B cells (CD45R^+^, IgG1^+^, GL7^−^, CD38^+^), and plasma cells (CD45R^−^, IgD^−^, CD138^+^) increased with the combined use of K3 compared to the scrambled control peptide.

Proliferation and differentiation of T- and B-cells were also evaluated in the spleen, which controls the systemic immune response. The total number of splenocytes significantly increased in the K3 combination group compared with the scrambled combination control group, and the increase of B cells was significant (Figure 4D). In the subpopulation of T cells, the combined use of K3 also increased effector T cells, effector memory T cells, and central memory T cells compared to the control group (Figure 4E). From these results, it was clear that K3 promoted the activation of naive T cells into effector T cells and was successful at inducing T cells with memory function. Furthermore, in B cells in the spleen, the populations of GC B cells, memory B cells, and plasma cells were increased by the combined use of K3 compared to the control, which had similar effects as those observed in DLNs from mice administered the K3 combination treatment (Figure 4F). Plasma cells that produced antibodies significantly increased with the combined use of K3, which have contributed to the higher antibody titer in this treatment group than in the control group. The GC that potentially formed due to an increase in effector T cells and GC B cells was evaluated by fluorescent immunostaining; however, no GC (PNA) fluorescence was detected and no K3-dependent effect was observed (data not shown). These results indicate that K3 is a TCI formulation adjuvant that not only excelled in antibody production, but also activated a local immune response in DLNs and a systemic immune response. Furthermore, because memory T- and B-cells were induced, these results suggest that this treatment would support the immune system to quickly respond to reinfection.

Antigen re-stimulation was performed in splenocytes after multiple immunizations and cytokine production in the culture supernatant was measured. We found that most IL-5, IL-9, IL-10, IL-13, IL-17, and TNF-α cytokines were below detectable levels in both K3 and scrambled control combination groups (Figure 4G). The IFN-γ and IL-2 cytokine production increased with combined K3 treatment compared to the control, suggesting that the K3 treatment could differentiate T_N_ into Th1 cells.

From these results, we show that administration of K3 at the skin promoted the activation of T- and B-cells and was successful in inducing their differentiation into memory and plasma cells, as well as inducing Th1-type immune responses. These results further indicate that K3 is a promising adjuvant for TCI formulations. 

### 3.4. Characteristics of the K3-Induced Primary Immune Response after Transcutaneous Vaccination

We analyzed the proliferation of OVA-specific CD4^+^ T cells (i.e., transferred OT-II cells) and OVA-specific CD8^+^ T cells (i.e., transferred OT-I cells) transcutaneously immunized with OVA and K3 or the scrambled control peptide. In DLNs, the division frequency with the K3 or scrambled control combination treatment was similar in both OT-II and OT-I cells, indicating that K3 did not promote proliferation (Figure 5A). Furthermore, the number of T cells 4 days after the first immunization was almost the same even when K3 was used in immunized WT mice (Figure 5B). These results show that when K3 is administered at the skin, it did not enhance the antigen presentation ability in LCs and dDCs that express TLR9 to T cells. However, the population ratio and number of B cells increased 4 days after the primary immunization (Figure 5C). The activated state of these proliferated B cells was evaluated using the cell surface marker, GL7, which is expressed in activated germinal center B cells; however, GL7 expression did not increase in when K3 was used in combination with OVA (Figure 5D), suggesting that the treatment had not yet led to the differentiation and activation of GC B cells. Expression of the early activation marker, CD69, increased in the K3 combination group compared to the control, indicating that B cells were in an activated state 4 days after immunization with K3. The expression of the costimulatory molecule, CD86, was increased by the combination of K3; however, CD80 expression showed no difference among the combination treatments of K3, PBS and scrambled control. These results were expected because B cells activated by the combined use of K3 could interact with T cells. Interestingly, the expression of MHC class II, which presents antigen peptides to T cells, was higher in both the K3 and the scrambled peptide control group than in the PBS group. This result suggests that TLR9 non-mediated stimulation can occur with non-specific nucleic acids. Therefore, when K3 is transcutaneously administered, it does not act on TLR9-expressing LCs and dDCs to increase the efficiency of antigen presentation to T cells, but rather promotes B cell activation and maturation.

To analyze whether transcutaneous administered K3 acts directly or indirectly on B cells, we first examined TLR9 expression in B cells. TLR9 expression was observed in the endosomal B cells and K3 detected in DLNs would act directly on B cells (Figure 5E). B cells isolated from WT or TLR9-KO mice were co-cultured with K3 or scrambled peptide, and B cell activation was evaluated. The co-culture was performed for only 6 h so differentiation into the GC B cells did not occur and GL7 expression did not increase as was observed in the in vivo test. The expression of other costimulatory molecules such as CD40, CD80 and CD86, and MHC class II significantly increased in WT mice by co-culture with K3. When K3 was added to B cells derived from TLR9-KO mice, the expression of these surface markers did not increase and was at a similar expression level as in the PBS and scrambled peptide control groups. These results suggested that K3 was transported into DLNs and acted directly on B cells though TLR9 and promoted B cell activation and maturation. 

### 3.5. Human Safety Evaluation for Transcutaneous Administration of K3

To evaluate the safety of transcutaneous K3 administration, a clinical trial using K3-loaded sdMN was conducted (Figure 6). In-patient identification number (ID)-1 and ID-2, the placebo-sdMN was applied and redness at the application site on day 1 (1 day after the first application) was reported. The inflammation at the application site completely healed by day 21 (3 weeks after first application). When placebo-sdMN were reapplied to another area of ID-1 and ID-2, the local inflammation on day 22 (1 day after the second application) was reduced compared to the first application. Additionally, no inflammation at the application site was observed on day 42 (3 weeks after second application). Conversely, four subjects who received K3-loaded sdMN application showed erythema with infiltration (ID-3) and inflammation (ID-4, ID-5, ID-6) on day 1. In ID-3, ID-4, and ID-5, pigmentation was observed at the application site on d 21; therefore, it was hypothesized that a strong inflammatory response to the sdMN application had occurred. After the second application, ID-3 showed a reduced local reaction on day 22 compared to the first application. In ID-4, the same level of inflammation as the first application was reported, and no local reaction was observed in ID-6. On day 42, these three subjects had no trace or slight pigmentation at the application site. In subject ID-5, inflammation on day 22 was stronger than at the time of the first application, and the application site was infiltrated on day 42. There were no systemic adverse or significant changes in vital signs, blood biochemistry, and urine in either the placebo-sdMN or the K3-loaded sdMN throughout the study (Appendix A). Based on these results, although further studies such as K3 dose optimization and sdMN length may be necessary to reduce local skin reactions, the K3-loaded sdMN were safely applied to human skin without inducing serious side reactions.

## 4. Discussion

Intradermal immunization (ID) with the hepatitis B vaccine, influenza vaccine, or Bacille de Calmette et Guérin has been reported to be superior to immune induction and dose sparing compared to subcutaneous immunization (SC) or intramuscular immunization [22,23]. We also confirmed that virus-derived components or adjuvants could enhance the specific antibody titer against purified protein antigen in ID more effectively than SC (Appendix A). Thus, TCI using MN that target skin tissue as well as ID was a vaccination route in which the adjuvant significantly contributed to the dose-sparing effect. To promote practical application of the sdMN formulation, we searched for promising adjuvants that could be applied to sdMN and had a potential for antigen-specific immune enhancement. In this study, we used the poke-and-patch method that allowed for TCI like sdMN because it was expensive and time-consuming to determine the conditions for antigens and various TLR ligands loading into the sdMN and to produce prototypes.

We found that the K3 was the most useful adjuvant identified from our search for candidate adjuvants. The combined use of K3 not only produced specific antibody production, but also successfully differentiated T cells and B cells into memory cells and plasma cells. K3 may have a sufficient vaccine effect even in elderly people whose immune cell functions have been reduced [24]. The results of IgG subclass and cytokine production revealed that the combined use of K3 could induce a Th1-type immune response. These immune responses lead to the production of highly neutralizing antibodies that eliminate pathogens and are very important factors in vaccine development. Additionally, the production of specific IgE antibodies that is a side reaction of vaccination was the same as that observed in the control conditions. Since K3 has these characteristics without severe side effects, we conclude that a transcutaneous vaccine combined with K3 may also be useful for vaccinating infants. When an influenza vaccine is administered to infants, the Th2-type immune response is induced and specific IgE antibodies are produced, which can lead to anaphylactic shock [25]. Thus, a transcutaneous vaccine combined with K3 could minimize side reactions and provide painless and highly effective vaccination even in infants.

We evaluated the induction of cellular immunity by measuring Th1-type immune responses with K3 and the induction of mucosal immunity as a biological defense system at the site of initial infection. However, no induction of cellular or mucosal immunity was observed in the adjuvant combination groups we tested. There are reports of transcutaneous vaccine cases in which cellular and mucosal immunity was induced, but most of these cases used exotoxins such as cholera toxin or enterotoxin [26]. The adjuvant activity mechanisms of these exotoxins are still unknown and there is a risk in their clinical application because they can elicit an antigen-non-specific immune response. If mucosal and cellular immunity can be induced in addition to specific antibody production by using TCI formulations with TLR ligands, these become ideal vaccine formulations that induce both systemic immunity protection and prevent pathogen invasion at mucosal surfaces. Therefore, we are currently optimizing the dose of TLR ligands and analyzing the immune response of sdMN loaded with TLR ligands to investigate whether they can induce cellular and mucosal immunity.

Regarding the mechanism of K3 adjuvant activity, we report that K3 exerted an effect in a TLR9-dependent manner. The expression of TLR9 was observed in LCs and dDCs present in the skin, suggesting that these antigen-presenting cells might be uniquely involved in the induction of immune responses. However, it was also found that (1) the division and proliferation of antigen-specific T cells was equivalent to that of the scrambled control group, (2) K3 quickly migrated to the DLNs, and (3) B cells in the DLNs were activated and proliferated. Thus, K3 acted on TLR9 expressed in B cells rather than acting on antigen-presenting cells during transcutaneous administration with the poke-and-patch method. Furthermore, we confirmed that the antigen-specific antibody production induced by the transdermal vaccine was T cell-dependent, which is similar to the conventional injection immunization method, suggesting that transcutaneously administered K3 exerted an adjuvant activity on T cell-dependent antigen-specific antibody production (Appendix A). The activation of B cells by K3 and the activation of T cells by transcutaneous administration induced a strong interaction, thus inducing the differentiation of immune cells. These results indicate that K3 exhibited adjuvant activity. In general, CpG-ODN has type A, B and C and differs in the promotion of activated cells and the produced cytokines [27]. Type A acts on antigen-presenting cells called plasmacytoid DCs and promotes IFN-γ production, while type B, which contains K3, acts on B cells and promotes IL-6 production by activating the NF-κB signal pathway [28,29]. Type C is thought to have both type A and type B properties [30]. Therefore, while the action of K3 was similar to conventional injection administration, it was efficiently transported into the DLNs from the skin surface layer, potentially through lymphatic vessels, where it acted on B cells in the DLNs.

In the clinical study, a local skin reaction occurred just by applying K3-loaded sdMN. In TCI, high concentrations of antigens and adjuvants are administered to the shallow part of the skin, making it easier to detect as a local reaction. Side reactions are likely to occur in correlation with excellent immune responses, and there is a possibility that side reactions occur more when loaded with vaccine antigens. Therefore, it is necessary to optimize the loading amount of K3 and the length and number of needles. The K3 is a CpG-ODN that also reacts with human TLR9, which is candidate for practical use as an adjuvant for TCI formulations. The K3 production system under GMP standards have already been established and its safety and efficacy by injection administration has been shown in clinical trials for a malaria vaccine combined with K3 [31,32]. We will optimize the TCI formulations in addition to its efficacy in humans and work toward its practical use as a component of safe and effective transcutaneous vaccines.

In this study, we found that K3 is a promising adjuvant for transcutaneous vaccine that reduces the amount of antigen and the number of immunizations required to elicit an immunity-inducing effect. The addition of K3 as a vaccine adjuvant can induce a Th1-type immune response and K3-loaded MN can be safely applied to human skin, suggesting that TCI formulations that include K3 support safe and effective vaccination programs.

## Figures and Tables

**Figure 1 pharmaceutics-12-00267-f001:**
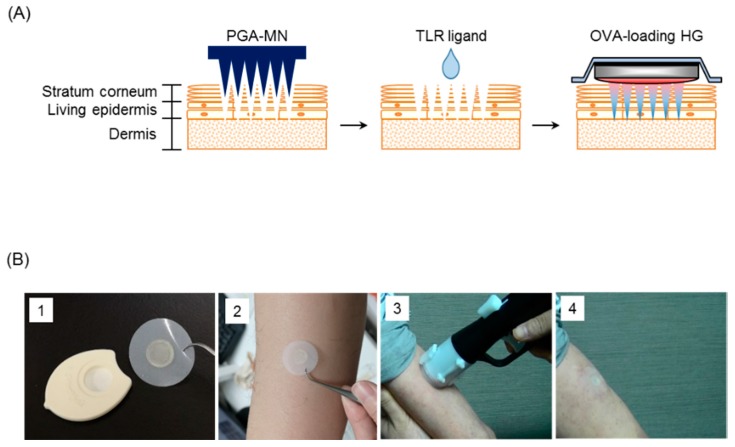
Methods for epicutaneous immunization in mice and humans. Puncture holes were made on the hairless back skin of mice using biodegradable polyglycolic acid-microneedle (PGA-MN). A 5-μL volume of toll-like receptor (TLR) ligand and OVA-loaded hydrophilic gel patch (HG) were applied to the holes (**A**). In human subjects, (1) K3 (200 μg)-loading self-dissolving microneedle (sdMN) and placebo-sdMN were fixed to the plastic case that (2) was placed on the skin of the lateral upper arm and (3) applied with a handheld spring-type applicator that was (4) attached for an hour (**B**).

**Figure 2 pharmaceutics-12-00267-f002:**
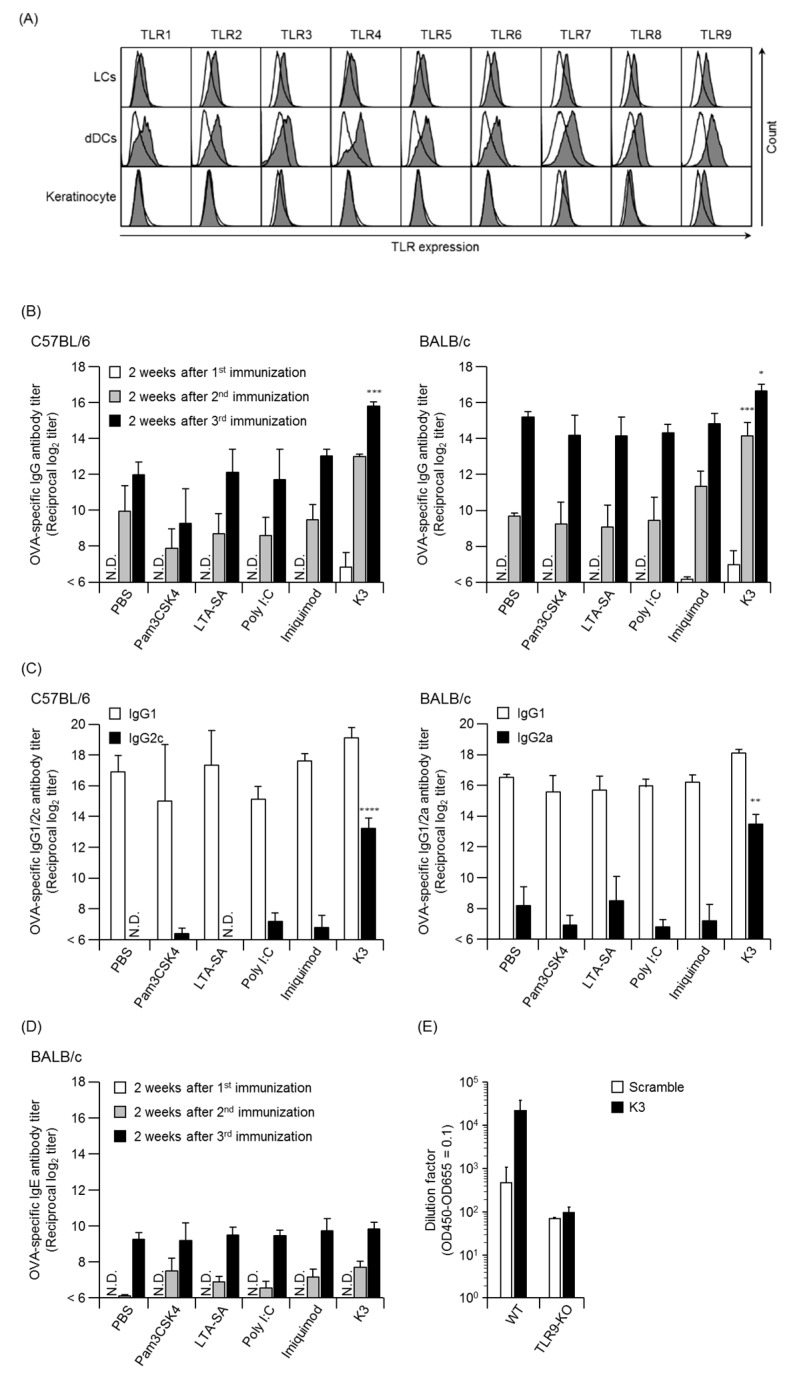
Expression analysis of TLRs in skin cells and the OVA-specific antibody response with the poke-and-patch method. Epidermal and dermal cell suspensions were analyzed for TLR expression by flow cytometry (**A**). The open histogram represents the isotype control group and the filled histogram represents the TLR staining group. C57BL/6 and BALB/c mice were immunized using the poke-and-patch method three times at 2 w intervals with OVA in combination with each TLR ligands. Sera were collected after immunization and assayed to determine the antigen-specific IgG (**B**), IgG subclass (**C**), and IgE (**D**) titers by ELISA. WT mice or TLR9-KO mice were epicutaneously immunized with K3 or the scramble peptide using the poke-and-patch method. Two weeks after the third immunization, OVA-specific IgG antibody titers were measured by ELISA (**E**). Data are expressed as mean ± SEM of results from three to five mice. (Student’s *t*-Test, * *p* < 0.05, ** *p* < 0.01, *** *p* < 0.005, **** *p* < 0.001).

**Figure 3 pharmaceutics-12-00267-f003:**
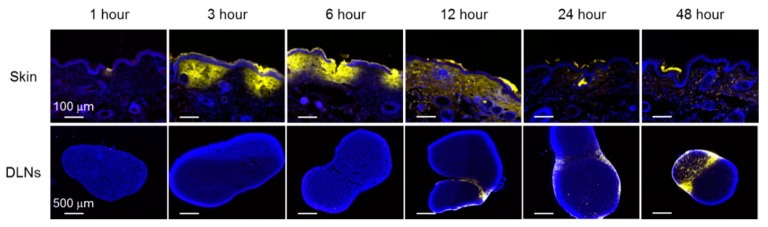
Localization of K3 for transcutaneous formulation. Puncture holes were made on the hairless back skin of HR-1 mice using PGA-MN. A 20 μg/5 μL AF647-labeled K3 solution and 10 μg OVA was loaded on HG that was then attached to the puncture site. The treated skin and DLNs were collected and frozen in liquid nitrogen. Frozen sections were stained and observed for OVA (red), K3 (yellow) and nucleus (blue) by fluorescence microscopy.

**Figure 4 pharmaceutics-12-00267-f004:**
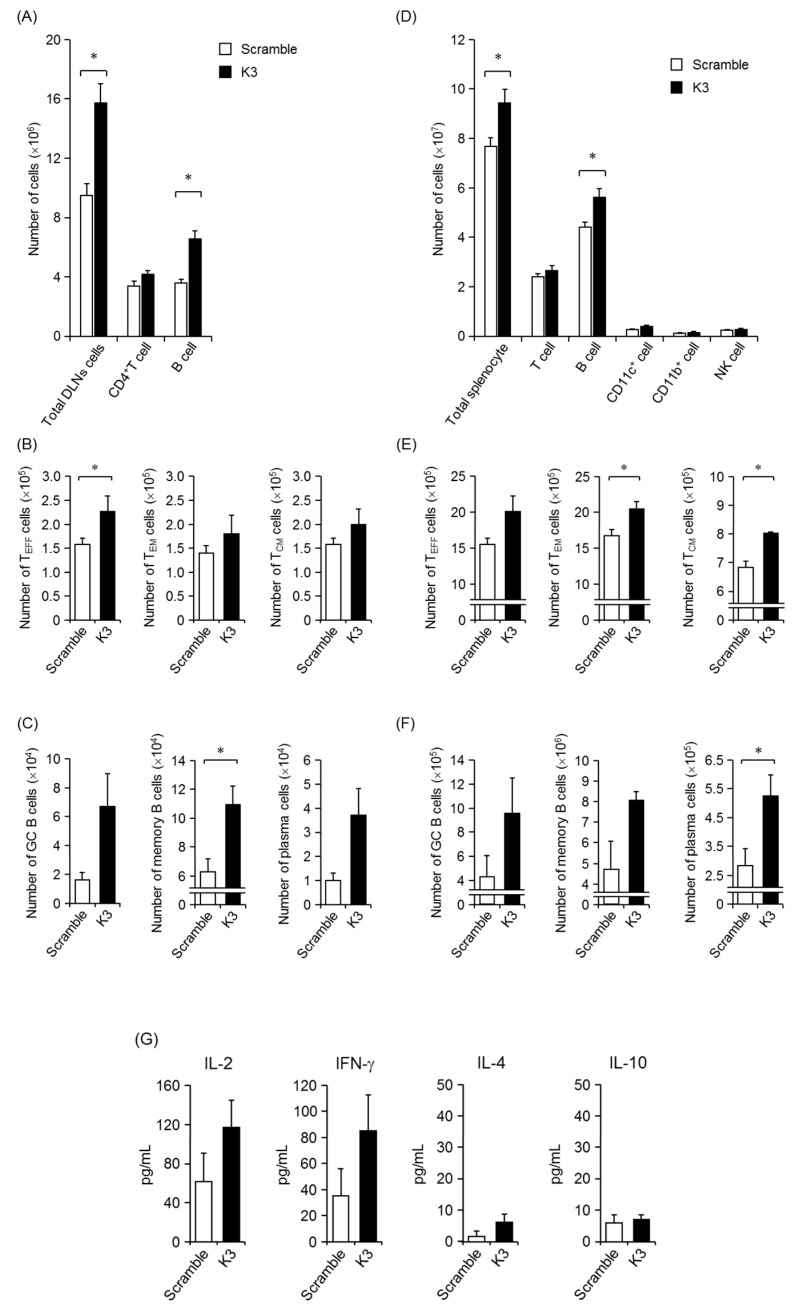
Activation and differentiation of T- and B-cells in combination with K3. C57BL/6 mice were epicutaneously immunized three times at 2-week intervals with K3 or the scramble control using the poke-and-patch method. The DLNs and spleens were collected 2 weeks after the final immunization; the number and differentiation of each immune cell was analyzed by flow cytometry (**A**,**D**). Activated T cells in the DLNs and spleens were analyzed using CD44 expression, and then fractionated into T_EFF_, T_EM_, or T_CM_ populations by surface marker expression (**B**,**E**). The DLNs and spleens were stained for GC B cells, memory B cells and plasma cells (**C**,**F**). The splenocytes were prepared and cultured with 1 mg OVA for 48 h (**G**). The supernatant from these cultures was collected and cytokine production was evaluated with a Bio-Plex. Data are expressed as mean ± SEM of results from three mice. Data are expressed as mean ± SEM of results from three to five mice. (Student’s *t*-Test, * *p* < 0.05).

**Figure 5 pharmaceutics-12-00267-f005:**
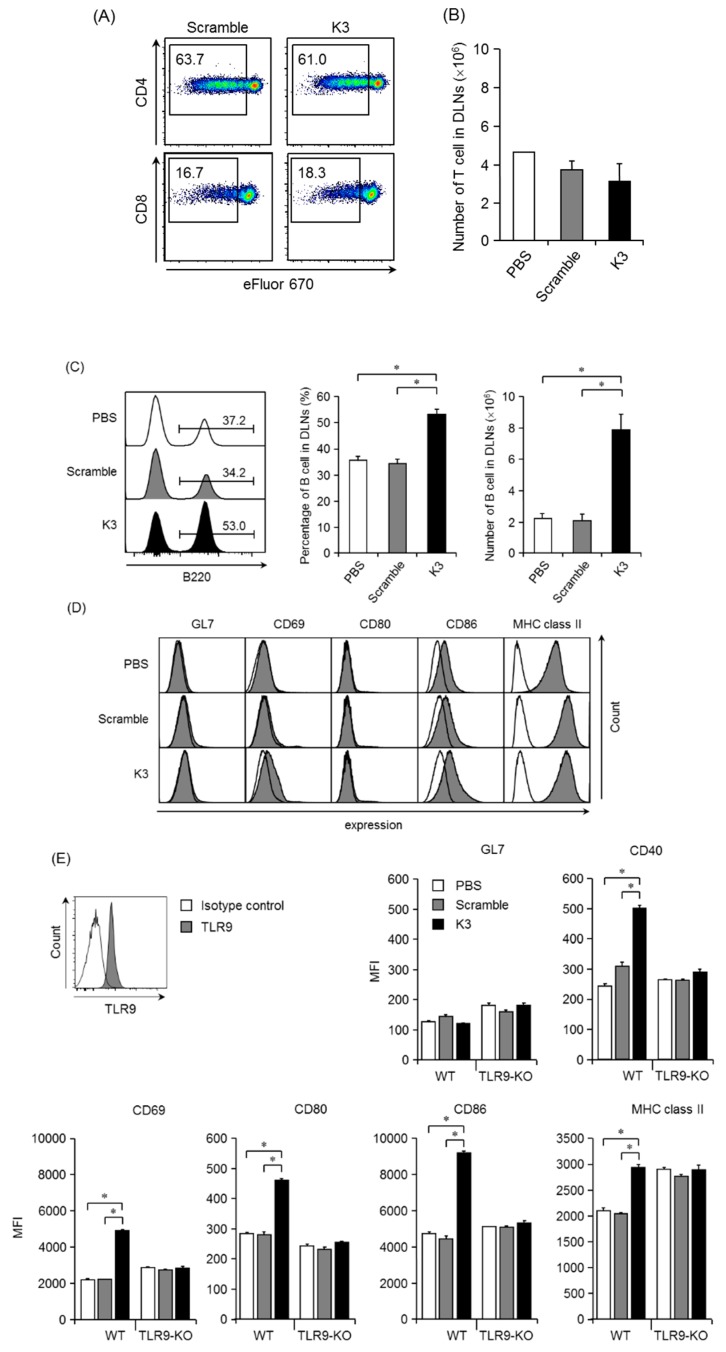
Characteristics of K3 in a transcutaneous vaccine formulation. WT mice (CD45.2, CD90.2) were intravenously transferred with eFluor 670-labelled OT-I (CD45.2, CD90.1) or OT-II cells (CD45.1, CD90.2). The next day, mice were epicutaneously immunized with K3 or the scramble control, and the cell proliferation of OT-I or OT-II cells from DLNs was analyzed by flow cytometry 4 days after immunization (**A**). The number of T- and B-cells in DLNs were analyzed by flow cytometry 4 days after immunization (**B**,**C**). The activation state of B cells was investigated with surface markers (**D**). TLR9 expression in B cells was analyzed by intracellular staining. B cells were isolated from WT or TLR9-KO mice and cultured for 6 h in the presence of K3 or the scramble control. The activated state of the B cells was analyzed with a surface marker (**E**). Data are expressed as mean ± SEM of results from three to five mice. (Student’s *t*-Test, * *p* < 0.05).

**Figure 6 pharmaceutics-12-00267-f006:**
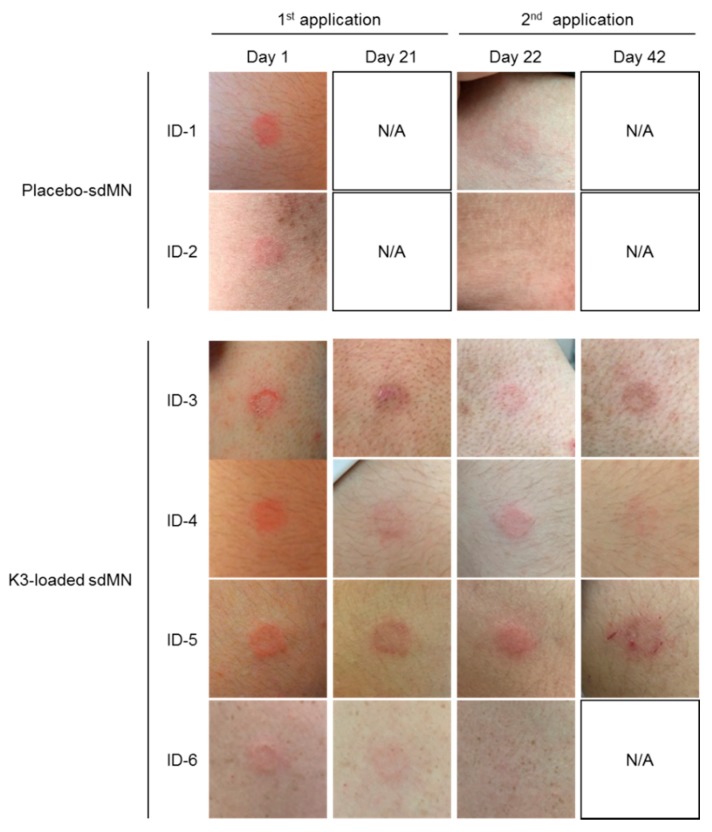
Safety evaluation of K3-loaded sdMN use in humans. Placebo-sdMN or K3-loaded sdMN were applied to the upper arm of participants twice every 3 weeks using an applicator. The day after administration (Day 1, Day 22) and 3 weeks after the first application (Day 21, Day 42); local skin reactions at the application site were observed.

**Table 1 pharmaceutics-12-00267-t001:** Types and description of TLR ligands using transcutaneous administration.

TLR Ligand	Description	Molecular Weight	Dose	Target TLR
Pam3CSK4	Synthetic mimic of bacterial lipoprotein	1500 Da	20 μg	TLR1/2
LTA-SA	Lipoteichoic acid from *Staphylococcus aureus*	4000–8000 Da	100 μg	TLR2
Poly I:C	Synthetic analogue of double stranded RNA	1500–8000 Da	50 μg	TLR3
Imiquimod	Small molecule agonist	240 Da	100 μg	TLR7
K3	CpG-oligodeoxyribonucleotide (ODN)	6300 Da	20 μg	TLR9

**Table 2 pharmaceutics-12-00267-t002:** The clinical protocol for assessing the safety of the K3-loaded sdMN.

Day	0	1	2–7	8–20	21	22	23–28	29–41	42
Vaccination	●								
Medical examination	●	●			●	●			●
Vital sign measurement	●	●			●	●			●
Blood count and biochemistry	●	●			●	●			●
Urinalysis	●	●			●	●			●
Cytokine production	●	●			●	●			●
Diary by subjects	●	●	●	▲	●	●	●	▲	▲
Confirmation of adverse events			●				●		

● Required; ▲ When an adverse event occurs.

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
