# Peer review of "Characteristic of K3 (CpG-ODN) as a Transcutaneous Vaccine Formulation Adjuvant"

_pharmaceutics, 2020, doi:10.3390/pharmaceutics12030267_

Round 1

Reviewer 1 Report

Attached as file below.

A.

Reviewer 2 Report

Manuscript ID: Pharmaceutics-737068

Title: Characteristic of K3 (CpG ODN) as a transcutaneous 2 vaccine formulation adjuvant

Recommendation: Publish after minor revision

This manuscript from Sayami Ito et al. describes the use of K3 (CpG oligonucleotide ), a ligand for Toll-like receptor 9 (TLR9) as a potential vaccine adjuvant in transcutaneous immunization.  Authors used biodegradable polyglycolic acid microneedles in combination with K3 and ovalbumin on the skin surface via poke and patch method. Initial study included screening of various TLR ligands in inducing Ovalbumin specific antibody responses. Further, authors carried various immune response studies using T & B cells isolated from spleens and DLNs.  In addition, authors performed a safety study using human subjects to demonstrate the effect of K3 loaded self dissolving microneedles on human skin. Overall, Authors showed the vaccine adjuvant effect of K3 in inducing Th1 immune responses.  All the experiments are  well-designed and executed.  Manuscript is well-written and clearly presented. I do recommend this work for publication after addressing the following concerns.

Concerns:

  1. Page 2&3, Lines 70-72: This sentence is not clear and needs to be corrected.

  1. Page 3, Line 79: It should be 'insolubility' instead of 'insoluble'.

  1. Page 3, Lines 86-88: This part needs improvement. A brief information of the work carried should be included.

  1. Table 1: Authors showed 'origin/type' of TLR ligands under 'property' section which needs a correction. For example, chemical name was included under 'property' for Imiquimod.     

  1. Page 8, Section 2.11: I do not see the dosage information of K3 loaded MN used for safety study. Authors must include the amount used for this study.  

  1. Page 11, Lines 323-329: Authors used TLR9-KO mice to demonstrate the selective adjuvant activity of K3 through TLR9 receptors. It is known that K3 is a TLR9 ligand. However, it is not clear how authors concluded that the adjuvant activity is selective via TLR9 without performing similar study in other  TLR-KO mice.  Authors must explain.

  1. Figure 4: Figure 4A and 4B labels are missing in the Figure legend.

  1. Page 25, Lines 494-499: Authors claim that there were no adverse effects of K3-MN on human subjects. However,  it would be good to include the data obtained from patients as supporting information to strengthen their statement.  

  1. Stability is an important issue for vaccine adjuvants. Why the stability study was not performed in this work.  Authors should explain on the stability of K3 vaccine adjuvant.

  1. K3 is a 'B-class' TLR9 ligand (K-type ODN) and it triggers NF-kB signals by stimulating B-cells via TLR9 activation. Activation of NF-kB signal pathway increases IL-6 & TNF-α production. Authors measured several cytokines including  IL-6 & TNF-α.  However, TNF-α expression levels were below detectable level.  Please explain.      
